# Current Sex Distribution of Cooking and Food Shopping Responsibilities in the United States: A Cross-Sectional Study

**DOI:** 10.3390/foods11182840

**Published:** 2022-09-14

**Authors:** Maximilian Andreas Storz, Kai Beckschulte, Maria Brommer, Mauro Lombardo

**Affiliations:** 1Center for Complementary Medicine, Department of Internal Medicine II, Freiburg University Hospital, Faculty of Medicine, University of Freiburg, 79106 Freiburg, Germany; 2Interdisciplinary Medical Intensive Care (IMIT), Medical Center—University of Freiburg, Faculty of Medicine, University of Freiburg, 79106 Freiburg, Germany; 3Department of Human Sciences and Promotion of the Quality of Life, San Raffaele Roma Open University, 00166 Rome, Italy

**Keywords:** home cooking, meal preparation, food shopping, household responsibilities, public health nutrition, food behavior

## Abstract

Home cooking is an important obesity prevention strategy and associated with benefits for diet and health. Although cooking may be a joyful act of mindfulness, it also requires planning, preparation and time. Historically, women have been more likely to fulfill the role of food shopping and cooking. More recent studies suggested a transition in traditional household role assignments towards a larger involvement of males. This study examined the current sex distribution of cooking and food shopping responsibilities in the United States of America based on a nationally representative sample of 9078 citizens from the National Health and Nutrition Examination Surveys (2017–2020). More than 80% of women aged 25 years or older indicated that they were the person who did most of the planning/preparing of meals in their families, whereas the percentage of males responding affirmatively was substantially lower, ranging from 38.73% to 43.20% depending on age. Analyses on food shopping duties revealed a comparable distribution. In multivariate regression, female sex was associated with significantly higher odds of being the main food shopper/meal preparer in the family (OR: 4.82 (4.14–5.60) and 5.54 (4.60–6.67), respectively). Our data suggest that the majority of food shopping and cooking duties are still performed by women, which has important implications for public health nutrition initiatives.

## 1. Introduction

Preparing healthy, diversified and tasteful meals for a balanced diet over a longer period requires time, patience, perseverance, creativity, dedication and cooking skills [1,2]. This is especially the case when cooking needs to be carried out for several people with varying preferred food choices, for example, in larger families or with family members adhering to special diets [3,4]. Cooking is usually an unpaid task that has been historically assigned to women at the expense of market work and private life [5,6,7].

In contrast to market work which yields a certain wage, the value of cooking is difficult to measure. Cooking plays an important role in the physical and mental well-being of all involved individuals [8,9]. Home cooking has also been associated with healthier dietary patterns, and potentially reduces the likelihood of nutrition-related disorders [10,11], whereas the frequent consumption of meals prepared away from home has been associated with nutritional deficiencies and an increased risk of all-cause mortality [12,13,14].

Traditionally, women have been more likely to fulfill the role of feeding the family in the United States [3,15]. That particular role comprised several time-consuming activities, such as meal planning, food shopping and food preparation [16,17]. With movements demanding a more equal treatment of women and men [18], increasing attention has been paid to the sex gap in unpaid household work and food shopping and cooking duties in particular [19,20]. A Canadian analysis suggested that men’s share of cooking has risen substantially, as they now do about 40 percent of all cooking [21]. Younger generations in particular have apparently moved away from the traditional division of historical roles, and attempt to better balance the work related to cooking and shopping activities [1,22]. Notably, some authors also reported opposite findings [23], and suggested that although younger generations have become much more open-minded about gender roles, those aged 18 to 34 were no more likely than older couples to divide most household chores equitably [24,25].

Taillie investigated potential genders gaps with regard to cooking activities, meal preparation and food shopping for meals [26]. Using data from the American Time Use Study (2003–2016) [27], this study suggested that home cooking generally increased in the aforementioned time span. College-educated men in particular showed the greatest increase in cooking activities, as their percentage increased from 37.9% in 2003 to 51.9% in 2016. Notably, the proportion of cooking men with less than a high school degree did not change significantly. In women, cooking frequency increased only slightly (from 64.7% to 68.7% for college-educated women). Although cross-sectional, these data suggest that a potential change is taking place, and that historical household role assignments might be subject to transition.

Yet, follow-up data to this investigation are scarce, and other US-based data sources have been rarely investigated with regard to that particular question. To gain deeper insights into the current sex distribution of cooking and food shopping responsibilities in US households, we examined data from the US-based National Health and Nutrition Examination Surveys (NHANES (2017–2020)) [28].

More specifically, we aimed to investigate the following hypotheses:(a)Sex gaps in cooking duties and food shopping based on traditional role models are still present in the United States but potentially less pronounced.(b)Cooking and food shopping duties are more equally distributed in younger generations, as opposed to older generations.

## 2. Materials and Methods

### 2.1. The NHANES

To investigate the aforementioned hypotheses, we used data from the NHANES [28]. The NHANES is a nationally representative program of studies run by the National Center for Health Statistics (NCHS) designed to assess the health and nutritional status of adults and children in the United States of America. NHANES is an ongoing survey examining a representative sample of about 5000 people per year across the United States. Survey data from the NHANES are frequently used by healthcare professionals to determine the prevalence of major diseases and their risk factors in the United States.

The NHANES has two major components: the interview component and the examination component. The interview component contains demographic, socioeconomic, dietary, and health-related questions. The examination component includes laboratory tests and other medical, dental, and physiological measurements administered by highly trained medical personnel. For additional details, we refer the reader to the NHANES overview brochure summarizing key features of the survey [29].

For this particular analysis, we used data from various NHANES modules, including demographic data and questionnaire data. All data stem from the NHANES 2017–2020 pre-pandemic cycle [30]. As of March 2020, the NHANES program has suspended their field operations due to the coronavirus disease 2019 (COVID-19) pandemic. As a result, data collection for the NHANES 2019–2020 cycle was not completed at this point, which implies that the collected data were not nationally representative. Therefore, NHANES combined all data collected from 2019 to March 2020 with data from the NHANES 2017–2018 cycle with the aim of forming a nationally representative sample of NHANES 2017-March 2020 pre-pandemic data.

### 2.2. Demographic Data

Demographic data included age, sex (male and female), race/ethnicity (Mexican American, Other Hispanic, Non-Hispanic White, Non-Hispanic Black, Non-Hispanic Asian, and Other Race—including Multiracial), marital status (married/living with partner, widowed/divorced/separated, never married), educational level (less than 9th grade, 9–11th grade, high school graduate/GED or equivalent, some college or AA degree, college graduate or above), and ratio of family income to poverty. For all items, we used the pre-defined NHANES categories without performing particular modifications. Individuals were included regardless of household size, since the NHANES 2017-March 2020 pre-pandemic cycle did not include this variable.

### 2.3. Meal Planning/Preparing/Shopping Data

Included questionnaire data stem from the Diet Behavior and Nutrition (DBQ) questionnaire [31]. The DBQ contained personal interview data on various dietary behavior and nutrition-related topics. This module included information on participants’ meal planner/shopper/preparer status at home. Questions included: “Are you the person who does most of the planning or preparing of meals in your family?” and “Are you the person who does most of the shopping for food in your family?”. Answer categories included “yes”, “no”, and “don’t know”. Only participants who gave a definite answer (yes/no) were considered in the final analysis.

The module also inquired whether participants shared planning or preparing of meals and shopping duties with someone else. The same answer categories applied here. Data from the DBQ formed the cornerstone of our analysis. Additional information may be obtained from the DBQ module description [31].

### 2.4. Statistics

Stata version 14 (StataCorp., College Stadion, TX, USA) was used for the statistical analysis. Appropriate sample weights provided by the NHANES were used to account for the complex, multistage, probability sampling design of the NHANES. All continuous variables were described with their mean and corresponding standard error in parentheses. Categorical variables were described as weighted proportions with their corresponding standard error.

To allow for statistically valid population inferences from sample data, we computed standard errors using established survey data procedures (including Taylor series linearization) that took into account the complex nature of the sample design. Reliability of the presented weighted proportions was assessed based on the current reporting standards for proportions by the NCHS [32].

For categorical variables, we used STATA’s design-adjusted Rao–Scott test (a design-adjusted version of the Pearson chi-square test) to explore potential associations between sex and main meal planner/food shopper/food preparer status. In weighted subanalyses, we also stratified results by race/ethnicity and age category.

Furthermore, we ran multivariate logistic regression models to investigate potential associations between sex and main meal planner/food shopper/preparer status after adjusting for covariates. Covariates included age, ethnicity, marital status, education level, and ratio of family income to poverty. Potential covariates were chosen based on initial exploratory bivariate analyses and based on previous publications in the field. All tests were two-sided and statistical significance was determined at α = 0.05.

## 3. Results

The final sample for this study comprised *n* = 9078 individuals aged 18 years or older, which may be extrapolated to represent 237,664,985 US Americans. Almost 52% of participants were female (weighted proportion). The mean age of our sample was 48.39 (0.54) years (Table 1). More than 62% (weighted) of the sample were of Non-Hispanic White origin, and more than 11.3% (weighted) were Non-Hispanic Blacks.

Almost 62% of participants (weighted proportion) were married or lived with a partner. Approximately 27% had a high school degree, and more than 31% (weighted) had a college degree or higher. Table 1 also shows the ratio of family income to poverty, which was below 1 in more than 10% of the investigated sample (weighted).

### 3.1. Main Meal Planner/Preparer

As suggested by Figure 1, more women than men responded affirmatively to the question “Are you the person who does most of the planning or preparing of meals in family?”. Almost 80% of females replied with “yes” to that question, whereas only 40% of males agreed with that statement.

In a second step, we analyzed the replies to that particular question stratified by age category (Table 2). In females, the weighted proportion of participants replying with “yes” to that particular question increased by more than 30% from the first (18–24 years) to the second age category (25–34 years). The highest weighted proportion of males replying with “yes” was found in individuals aged 55–64 years.

### 3.2. Main Food Shopping Duty

A comparable picture was found with regard to the question “Are you the person who does most of the shopping for food in family?” (Figure 2). A significantly higher proportion of females stated that they were mainly responsible for food shopping (78.2%, weighted proportion). Fifty-eight percent of males denied being the person who did most of the shopping for food in their families.

An uneven distribution of this duty between males and females is also suggested by Table 3, in which the results are stratified by age category. The weighted proportion of males replying affirmatively to that particular question was highest in individuals aged 45–54 years (45.59%).

### 3.3. Analyses by Race/Ethnicity

We performed the same analyses stratified by race/ethnicity, as shown in Table 4 and Table 5. With regard to the first question, we observed a particularly uneven distribution in both sexes in Non-Hispanic Asians. Only 26.2% of males indicated that they were the person who did most of the planning/preparing of meals in their families. In comparison, 80.78% of females responded affirmatively to this statement. The highest weighted proportions of male participants responding affirmatively to this particular question were found in Non-Hispanic Blacks (43.81% (1.65)) and Other Race (44.73% (4.46)). In terms of food shopping duties, racial/ethnical differences were not as pronounced as compared to meal planning and preparation duties. Non-Hispanic Asian males most frequently responded affirmatively to this question (45.62% (3.07)), followed by Mexican Americans (43.97% (2.05)).

### 3.4. Shared Meal Planning/Preparing Duty

In addition to that, we explored whether participants shared the planning or preparing of meals with someone else (Figure 3). Compared with the previous questions, this analysis revealed a more balanced picture between both sexes.

### 3.5. Shared Food Shopping Duty

In addition to that, we investigated whether participants shared the shopping for food with someone else (Figure 4). Almost 62% of males replied “yes” to that question, whereas only 50.3% of females did so.

### 3.6. Logistic Regression Models

Finally, we used multivariate logistic regression models to ascertain the effects of sex on the likelihood of being mainly responsible for food shopping and meal preparation (Table 6). Female sex was associated with significantly higher odds of being the main food shopper in the family (OR: 4.82 (4.14–5.60), *p* < 0.001). Moreover, female sex was also associated with significantly increased odds of being the main meal preparer (OR: 5.54 (4.60–6.67), *p* < 0.001) after adjustment for covariates.

Margin plots were used to graph statistics from the fitted models shown above (Table 6). Figure 5 displays the marginal adjusted predictions for both sexes in all possible age categories. The largest difference with regard to food preparation duties between both sexes was found in individuals aged 18–24 years. A somewhat similar distribution was found with regard to shopping duties (Figure 5, lower panel).

## 4. Discussion

The present study sought to gain deeper insights into the current sex distribution of cooking and food shopping responsibilities in the United States.

Our data suggest that the majority of shopping and cooking duties are still performed by women. More than 80% of women aged 25 years or older responded affirmatively to the statement that they were the person who did most of the planning or preparing of meals in the family. The percentage of males responding affirmatively to this statement was substantially lower. A comparable picture was found with regard to shopping duties. In our employed logistic regression models, female sex was associated with significantly higher odds of being the main food shopper and meal preparer in the family (OR: 4.82 (4.14–5.60) and 5.54 (4.60–6.67), respectively).

Home cooking has been identified as a potent strategy to improve dietary intake and to prevent obesity [10,33]. Home cooking has also been associated with numerous social, cultural and emotional benefits, and gives families control over their food supplies [33,34]. Despite these benefits, home cooking frequency has demonstrably declined in the second half of the twentieth century [35]. One study demonstrated that total time spent on cooking in the United States declined by more than 30% from 1965 to 1995 [35].

One of the most frequent cited reasons for this development is the dual burden in women (e.g., balancing household work and job duties), who have traditionally been the predominant food shoppers and preparers in families [16,36]. Although cooking may be joyful at times, it is also filled with time pressures and requires preparation and planning. In a world where many households often depend on every adult family member working, sometimes even in multiple jobs and assignments with non-standard and unpredictable hours, it is immensely difficult to balance paid work and unpaid work at home [37].

A global analysis by Wolfson et al. revealed that cooking frequency varies considerably around the globe [38]. Their analysis suggested that women (median frequency 5 meals/week) generally cook more than men (median frequency 0 meals/week). On the other hand, there is also some evidence that the traditional kitchen roles begin to shift and that men take an increasing role in food preparation [21,35]. Using data from the NHANES (2017–2020), we therefore investigated the current distribution of cooking and shopping responsibilities among men in women in the United States.

Overall, our results suggest that women are substantially more engaged in food shopping and cooking. Approximately 4/5 of women ≥25 years responded affirmatively to the statement that they were the person who did most of the planning or preparing of meals in family. In contrast, the percentage of males responding affirmatively to this statement was substantially lower (Table 2). A comparable picture was found with regard to shopping duties.

Our data are thus essentially in line with prior studies from the United States, suggesting an unequal sex distribution in cooking and shopping duties [26,39,40]. Notably, a comparison between the respective studies is difficult and should be performed with great caution in light of the different data sources (e.g., data stem from the American Time Use Study and older NHANES cycles with different variables). Flagg et al., for example, analyzed 2007–2008 NHANES data and demonstrated that women were more likely to report being mainly responsible for both meal preparation/planning and shopping [39]. According to their analysis, women were also less likely to report having no responsibility for those tasks (in comparison to men). Our results support the ongoing gender gap with regard to cooking and shopping duties that appears to be existent in younger generations, as well. However, the reservation must be made that a 1:1 comparison between studies may not be possible due to differences in the variables’ structures and the employed regression models.

The herein presented data might be useful for policy makers, healthcare professionals, social workers and other authorities. Developing a clear understanding of who engages in food preparation at home and why is of paramount importance to allow for the tailoring of healthy eating and home cooking interventions [41]. Home cooking interventions demonstrated clear benefits [42,43], yet their practical implementation is often a challenge. In a modern world where human and financial resources for healthcare interventions are often scarce, our data might be of value to point out groups that could benefit in particular from such interventions.

### Strengths and Limitations

The present analysis draws upon a number of strengths but also has various weaknesses that warrant further discussion. A major strength is the large and nationally representative dataset from the NHANES. The sample comprised almost 9100 individuals, which may be extrapolated to represent 237,664,985 US Americans. Data from the NHANES is often used in epidemiological research, and the NHANES is a trusted longitudinal study and one of the largest of its kind. It is a reliable source designed to assess health and health behavior in US adults. Additional strengths of our manuscript include the various subanalyses (stratified analyses by gender and race/ethnicity) as well as the regression models that include a variety of confounders. Thus, we believe that our data allow for new and important insights into current food shopping and cooking duties distribution in the United States.

On the other hand, the reservation must be made that we present cross-sectional data, and no causal inference can be drawn from this type of dataset. Unlike Taillie [26], we did not perform a trend analysis and instead provided a rather descriptive picture limited to a relatively short time frame (2017–2020). While we present relatively new data, we also acknowledge that our analyses are built on data that stems from pre-COVID-19 times. Changes in eating, shopping and meal preparation patterns during the pandemic have been reported globally—an important aspect that has to be taken into account [44,45,46,47].

Moreover, we acknowledge that the analyzed items were mostly self-reported and may thus be subject to bias, such as recall bias or reporting bias. Finally, we acknowledge that our analysis would have benefited from adding additional socioeconomic predictors (e.g., adult employment status) describing our sample. Considering how some of these factors intersect with sex to influence food provisioning behaviors would provide more nuanced insights and possible targets for nutrition interventions and programming. Unfortunately, the addition of some variables would have dramatically reduced the overall sample size, whereas other variables (e.g., the existence of a person to help with household duties) were simply unavailable.

Yet, despite those limitations, we believe that our data allow for important insights into the distribution of meal preparation and shopping duties in the United States. Our data will be of importance for other researchers in the field as they allow for comparisons and trend analyses in said behaviors in the future. Finally, our analysis might be of high translational value for public health authorities and other stakeholders engaged in health promotion. Basic findings from our investigation may be taken into account when tailoring age- and gender-specific health promotion programs designed to promote a healthier lifestyle in Americans.

Our data might be used in future studies as a comparison and to reflect trends, yet future trials should also take the COVID-19 pandemic into account. Post-pandemic studies including a more detailed set of covariates (e.g., the geographical location (rural versus city-side) are required to evolve the understanding of cooking and shopping duties distribution in the United States. A more detailed exploration of the causes and drivers of inequities is also warranted.

## 5. Conclusions

Home cooking has been associated with healthier eating patterns and may thus serve as a relevant public health strategy to improve dietary intake and to prevent obesity. Cooking may be joyful at times, but it may also be filled with time pressures and requires adequate preparation and planning. Some studies suggested that historical household role assignments in this context might be subject to change, and that men increasingly engage in cooking activities. Our cross-sectional data, however, suggested that the majority of shopping and cooking duties are still performed by women. More than 80% of women aged 25 years or older responded affirmatively to the statement that they were the person who did most of the planning or preparing of meals in family, and female sex was associated with significantly higher odds of being the main shopper and meal preparer in the family. Our data may have important implications for public health nutrition initiatives and may help in tailoring cooking interventions for specific groups. Additional studies in this field are warranted to identify barriers to engaging in home cooking and to identify factors that contribute to the uneven gender distribution of cooking and shopping responsibilities in the United States.

## Figures and Tables

**Figure 1 foods-11-02840-f001:**
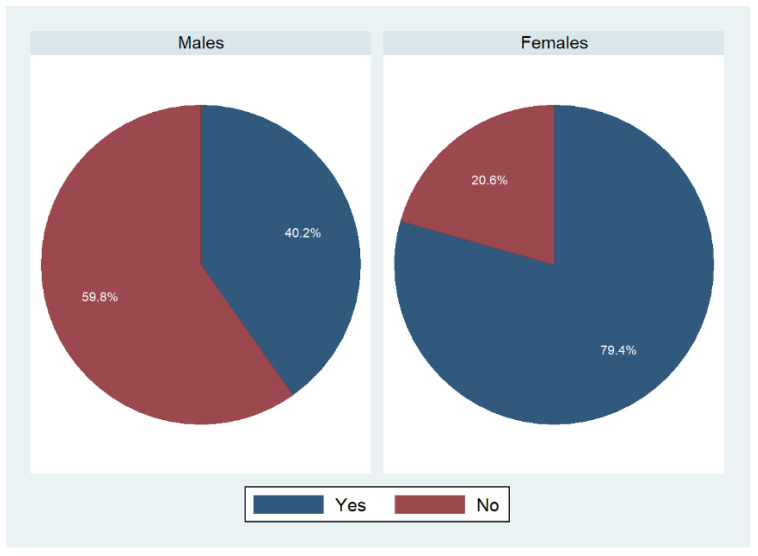
Meal planning and preparation in family stratified by gender: an overview. Weighted percentages, *n* = 9087. *p* < 0.001 based on a Rao–Scott test exploring potential associations between main person cooking and sex.

**Figure 2 foods-11-02840-f002:**
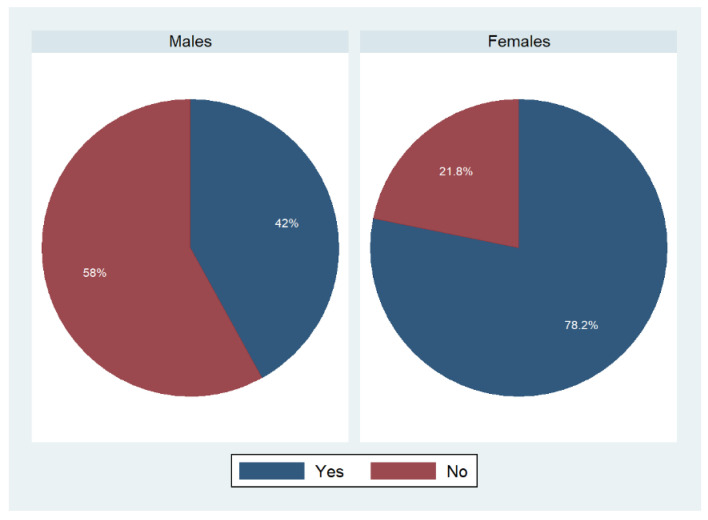
Shopping for food in families stratified by gender: an overview. Weighted proportions, *n* = 9087; *p* < 0.001 based on a Rao–Scott test exploring potential associations between main person shopping and sex.

**Figure 3 foods-11-02840-f003:**
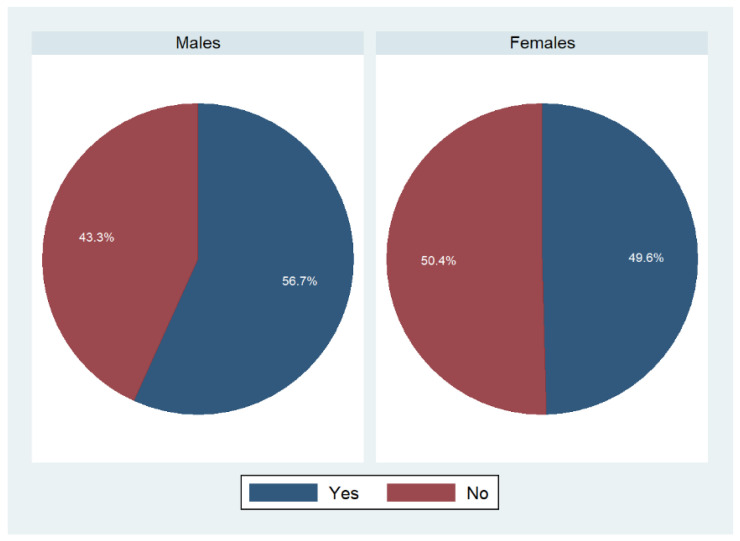
Sharing of meal planning and preparation duties stratified by gender: an overview. Weighted percentages, *n* = 9087; *p* < 0.001 based on a Rao–Scott test exploring potential associations between main person shopping and sex.

**Figure 4 foods-11-02840-f004:**
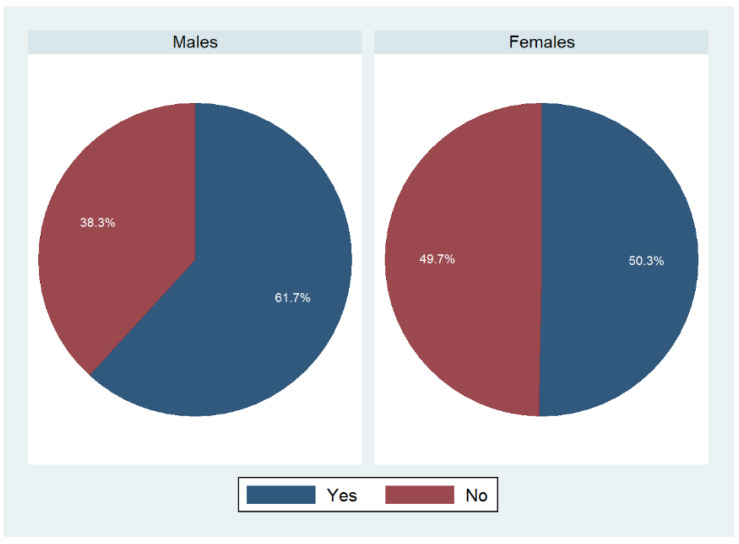
Sharing of food shopping duties stratified by gender: an overview. Weighted percentages, *n* = 9087; *p* < 0.001 based on a Rao–Scott test exploring potential associations between main person shopping and sex.

**Figure 5 foods-11-02840-f005:**
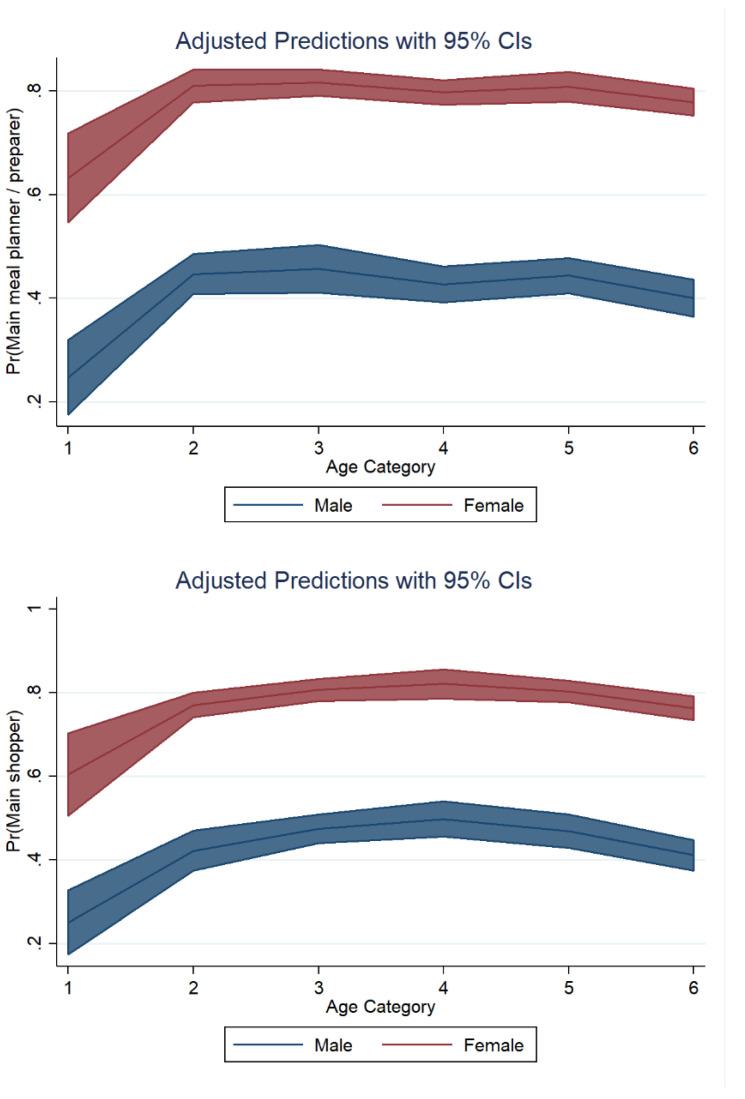
Main meal preparer responsibility and food shopping responsibility: marginal adjusted predictions. Marginal adjusted predictions for both sexes in all possible age categories. Upper panel: main meal preparer responsibility; lower panel: main shopping responsibility.

**Table 1 foods-11-02840-t001:** Sample characteristics (*n* = 9078).

**Age (years)**	
	48.39 (0.54)
**Sex**	
Male	48.09% (0.82)
Female	51.91% (0.82)
**Ethnicity/Race**	
Mexican American	8.34% (1.10)
Other Hispanic	7.54% (0.71)
Non-Hispanic White	62.81% (2.45)
Non-Hispanic Black	11.34% (1.40)
Non-Hispanic Asian	5.98% (0.93)
Other Race—Including Multiracial	3.99% (0.35)
**Marital status**	
Married/Living with Partner	61.95% (1.22)
Widowed/Divorced/Separated	18.80% (0.71)
Never married	19.25 (1.01)
**Education Level**	
Less than 9th grade	3.78% (0.37)
9–11th grade	7.15% (0.31)
High school graduate/GED or equivalent	26.87% (1.35)
Some college or AA degree	30.46% (0.90)
College graduate or above	31.75% (2.02)
**Ratio of family income to poverty**	
<1	10.58% (0.71)
≥1 and <2	16.85% (0.86)
≥2 and <3	13.39% (0.76)
≥3	59.17% (1.55)

Weighted proportions. Continuous variables shown as mean (standard error). Categorical variables shown as weighted proportion (standard error). All proportions can be considered reliable, as per recent NCHS Guidelines.

**Table 2 foods-11-02840-t002:** Participants’ replies to the question “Are you the person who does most of the planning or preparing of meals in family?” stratified by age category.

Age Category	Males		Females	
	Yes	No	Yes	No
18–24 years *	34.05% (4.87)	65.95% (4.87)	49.81% (5.66)	50.19% (5.66)
25–34 years **	42.45% (2.42)	57.55% (2.42)	80.20% (2.09)	19.80% (2.09)
35–44 years **	39.03% (2.84)	60.97% (2.84)	85.97% (1.66)	14.03% (1.66)
45–54 years **	41.20% (2.76)	58.80% (2.76)	80.37% (2.76)	19.63% (2.76)
55–64 years **	43.20% (3.50)	56.80% (3.50)	82.30% (2.36)	17.70% (2.36)
65 years and older **	38.73% (2.41)	61.27% (2.41)	82.26% (1.59)	17.74% (1.59)

Weighted proportions. Number of observations: 9078. All proportions can be considered reliable, as per recent NCHS Guidelines. * = *p* < 0.01; ** = *p* < 0.001.

**Table 3 foods-11-02840-t003:** Participants’ replies to the question “Are you the person who does most of the shopping for food in family?” stratified by age category.

Age Category	Males		Females	
	Yes	No	Yes	No
18–24 years *	33.05% (5.47)	66.95% (5.47)	49.56% (5.29)	50.44% (5.29)
25–34 years **	38.72% (3.45)	61.28% (3.45)	78.21% (2.34)	21.79% (2.34)
35–44 years **	40.14% (3.12)	59.86% (3.12)	85.74% (1.67)	14.26% (1.67)
45–54 years **	45.59% (3.45)	54.41% (3.45)	85.03% (2.47)	14.97% (2.47)
55–64 years **	44.79% (3.71)	55.21% (3.71)	81.82% (2.06)	18.18% (2.06)
65 years and older **	45.45% (2.39)	54.55% (2.39)	75.95% (2.06)	24.05% (2.06)

Weighted proportions. Number of observations: 9078. All proportions can be considered reliable, as per recent NCHS Guidelines. * = *p* < 0.01; ** = *p* < 0.001.

**Table 4 foods-11-02840-t004:** Participants’ replies to the question “Are you the person who does most of the planning or preparing of meals in family?” stratified by race/ethnicity.

Ethnicity	Males		Females	
	Yes	No	Yes	No
Mexican American *	38.46% (1.72)	61.54 (1.72)	83.09% (1.51)	16.92% (1.51)
Other Hispanic *	37.79% (3.11)	62.21 (3.11)	79.23% (3.08)	20.77% (3.08)
Non-Hispanic White *	41.14% (1.73)	58.86 (1.73)	79.10% (1.47)	20.90% (1.47)
Non-Hispanic Black *	43.81% (1.65)	56.19 (1.65)	78.57% (1.24)	21.43% (1.24)
Non-Hispanic Asian *	26.20% (2.37)	73.80 (2.37)	80.78% (1.84)	19.22% (1.84)
Other Race—Including Multiracial *	44.73% (4.46)	55.27 (4.46)	77.40% (3.75)	22.60% (3.75)

Weighted proportions. Number of observations: 9078. All proportions can be considered reliable, as per recent NCHS Guidelines. * = *p* < 0.001.

**Table 5 foods-11-02840-t005:** Participants’ replies to the question “Are you the person who does most of the shopping for food in family?” stratified by race/ethnicity.

Ethnicity	Males		Females	
	Yes	No	Yes	No
Mexican American *	43.97% (2.05)	56.03% (2.05)	80.18% (1.75)	19.82% (1.75)
Other Hispanic *	42.73% (3.20)	57.27% (3.20)	71.62% (3.10)	28.38% (3.10)
Non-Hispanic White *	41.15% (1.60)	58.85% (1.60)	79.31% (1.25)	20.69% (1.25)
Non-Hispanic Black *	43.34% (1.51)	56.66% (1.51)	77.73% (1.08)	22.27% (1.08)
Non-Hispanic Asian *	45.62% (3.07)	54.38% (3.07)	73.81% (2.04)	26.19% (2.04)
Other Race—Including Multiracial *	40.11% (3.90)	59.89% (3.90)	77.97% (4.42)	22.04% (4.42)

Weighted proportions. Number of observations: 9078. All proportions can be considered reliable, as per recent NCHS Guidelines. * = *p* < 0.001.

**Table 6 foods-11-02840-t006:** Logistic regression models to ascertain the effects of sex on the likelihood of being mainly responsible for shopping (main food shopping duty) and preparing meals (main planning/preparing) duty) after adjustment for selected covariates (age, ethnicity, marital status, education level and ratio of family income to poverty).

Age (years)	Main Food Shopper	*p*	Main Meal Preparer	*p*
18–24 years	-		-	
25–34 years	2.25 (1.45–3.48)	0.001	2.53 (1.64–3.93)	<0.001
35–44 years	2.80 (1.71–4.59)	<0.001	2.64 (1.68–4.16)	<0.001
45–54 years	3.08 (1.90–4.99)	<0.001	2.34 (1.50–3.64)	0.001
55–64 years	2.73 (1.75–4.27)	<0.001	2.50 (1.62–3.87)	<0.001
>65 year	4.14 (1.30–3.52)	0.004	2.08 (1.39–3.16)	0.001
**Sex**				
Male	-		-	
Female	4.82 (4.14–5.60)	<0.001	5.54 (4.60–6.67)	<0.001
**Ethnicity/Race**				
Mexican American	1.25 (1.03–1.51)	0.023	1.17 (1.03–1.32)	0.024
Other Hispanic	0.91 (0.73–1.14)	0.403	0.99 (0.82–1.21)	0.949
Non-Hispanic White	-		-	
Non-Hispanic Black	0.97 (0.85–1.11)	0.648	1.00 (0.88–1.12)	0.944
Non-Hispanic Asian	0.97 (0.82–1.16)	0.758	0.77 (0.65–0.91)	0.004
Other Race—Including Multiracial	0.86 (0.64–1.15)	0.294	0.97 (0.68–1.37)	0.843
**Marital status**				
Married/Living with Partner	0.85 (0.71–1.02)	0.085	0.94 (0.78–1.15)	0.551
Widowed/Divorced/Separated	2.08 (1.55–2.79)	<0.001	2.25 (1.61–3.13)	<0.001
Never married	-		-	
**Education Level**				
Less than 9th grade	-		-	
9–11th grade	0.96 (0.73–1.26)	0.756	1.22 (0.92–1.63)	0.165
High school graduate/GED or equivalent	1.18 (0.84–1.68)	0.325	1.38 (1.07–1.78)	0.014
Some college or AA degree	1.37 (0.99–1.90)	0.057	1.64 (1.28–2.09)	<0.001
College graduate or above	1.74 (1.26–2.41)	0.002	1.86 (1.36–2.53)	<0.001
**Ratio of family income to poverty**				
<1	-		-	
≥1 and <2	1.02 (0.80–1.30)	0.876	0.94 (0.74–1.19)	0.579
≥2 and <3	0.91 (0.71–1.17)	0.456	0.80 (0.60–1.06)	0.117
≥3	0.78 (0.32–0.98)	0.032	0.74 (0.61–0.89)	0.003

Both logistic regression models were statistically significant (F(20,6) = 22.13, *p* < 0.001 and F(20,6) = 36.90, *p* < 0.001, respectively). “-” denotes the reference category.

## Data Availability

Data are publicly available online (https://wwwn.cdc.gov/nchs/nhanes/Default.aspx; accessed on 25 August 2022). The datasets used and analyzed during the current study are available from the corresponding author upon reasonable request.

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
