# Peer review of "Current Sex Distribution of Cooking and Food Shopping Responsibilities in the United States: A Cross-Sectional Study"

_foods, 2022, doi:10.3390/foods11182840_

Round 1
Reviewer 1 Report
Dear author(s),
Thank you for giving me the opportunity to review your valuable manuscript.
The paper addresses a very interesting topic of sex distribution of cooking and food shopping responsibilities.
However, there are several issues to be addressed by the author:
· The theoretical background should be enriched. 35 references are not impressive for a WoS indexed journal.
· Line 57, 69: Please provide references for those surveys. Moreover, as the second one is the data source for this study, a reference should be provided in the methodological section.
· Line 70-73: each research hypothesis should be placed in a separate line. Moreover, for each of them, should be provided theoretical arguments and arguments based on the results of previous research.
· Usually, figures have a title, describing their content. Not the actually questions from the survey.
· Line 179: I suppose that In a second step, the replies were stratified by age category, not sex.
· Table 2 and so on …: Each table and figure should mentioned within the text before it is inserted.
· Line 214-226: only he distribution by rese for the first question was analyzed. Moreover, I expect more analysis from two tables of data.
· Line 263-264: Within the text you are referring to the models above, respectively the effect of sex on main meal shopper and planner. However, the title of figure 5 is referring to sharing those duties. Please make clear.
Author Response
Dear Reviewer,
We would like to thank you very much for careful and thorough reading of this manuscript and for the thoughtful comments and constructive suggestions, which help us to improve the quality of this article. Please kindly find our response below. All requested changes have been clearly marked in yellow and blue color. We appreciate your input, your advice and the fast peer review. Please find our point-by-point response below. Thank you!

Reviewer 2 Report
Dear authors
When analyzing the content of the manuscript, I found lot of confusing numbers. Please take into consideration all the comments raised.
Comments manuscript: foods-1909469
Brief statement:
The present study sought to gain deeper insights into the current sex distribution of 275 cooking and food shopping responsibilities in the United States. It is a good manuscript; however, it needs some edits:
Abstract:
I suggest to reformulate the sentences, honestly I didn’t feel the manuscript attractive when I red the abstract.
Line 124: the link of DBQ is not working. Also, is this questionnaire was validated before?
Results:
Please report your data as mean and SD.
What is the meaning of (0.82) in males and females?
Concerning the race and ethnicity: what are the meanings of the numbers put between parenthesis?
Also, these numbers shouldn’t be equal to 100?
Please reconsider analysis of this table. I guess there are errors in reporting the data.
In Figure 1. If you are reporting the data as yes and No by gender, thus try to focus on one option: either yes or no. Please change this figure to a pie for example showing the % of people saying No or Yes by gender.
In Table 2. Authors should pay attention to the calculation of the percentages. It should be equal to 100.
Figure 2. Same comment as figure 1
I guess all the tables should be recalculated.
Author Response

(The authors gave the same response as above.)

Round 2
Reviewer 2 Report
The manuscript shaped well